# From Digestion to Detoxification: Exploring Plant Metabolite Impacts on Insect Enzyme Systems for Enhanced Pest Control

**DOI:** 10.3390/insects16040392

**Published:** 2025-04-07

**Authors:** Masoud Chamani, MohammadReza Dadpour, Zahra Dehghanian, Sima Panahirad, Ali Chenari Bouket, Tomasz Oszako, Sumit Kumar

**Affiliations:** 1Department of Plant Protection, Faculty of Agriculture and Natural Resources, University of Mohaghegh Ardabili, Ardabil 56199-11367, Iran; 2Department of Horticultural Sciences, Faculty of Agriculture, University of Tabriz, Tabriz 51666-14838, Iran; dadpour.mr@gmail.com; 3Department of Biotechnology, Faculty of Agriculture, Azarbaijan Shahid Madani University, Tabriz 53751-71379, Iran; zdehganian@yahoo.com; 4Department of Horticultural Sciences and Landscape Engineering, Faculty of Agriculture, University of Tabriz, Tabriz 51666-14838, Iran; s.panahirad@tabrizu.ac.ir; 5East Azarbaijan Agricultural and Natural Resources Research and Education Centre, Plant Protection Research Department, Agricultural Research, Education and Extension Organization (AREEO), Tabriz 53551-79854, Iran; a.chenari@areeo.ac.ir; 6Department of Forest Protection, Forest Research Institute in Sekocin Stary, 05-090 Raszyn, Poland; t.oszako@ibles.waw.pl; 7KVK, Mau, Acharya Narendra Deva University of Agriculture and Technology, Ayodhya 224229, UP, India; sumit.kumar10@bhu.ac.in

**Keywords:** plant metabolites, co-evolution, oxidative stress, enzyme modulation, sustainable agriculture

## Abstract

Plants and insects are in a constant battle—plants use chemicals to defend themselves, and insects use enzymes to fight back and digest their food. Some enzymes help with digestion, while others protect insects from plant toxins. This review explains how these tiny battles work and how factors like pollution can affect them. Scientists hope to use this knowledge to develop safer, natural ways to control pests by targeting these enzymes, leading to more sustainable farming.

## 1. Introduction

Plants have developed sophisticated defense mechanisms against herbivorous insects, involving the synthesis of a wide range of secondary metabolites. These metabolites serve as chemical deterrents, making the plant unattractive or even harmful to potential pests. In response, insects have evolved diverse tactics to counteract these plant defenses and exploit the nutritional resources provided by their hosts. Insects employ a prominent strategy that involves manipulating digestive enzymes within their midguts to effectively process and extract nutrients from plant tissues [1,2].

Insects possess a multitude of digestive enzymes, such as proteases, lipases, amylases, sucrases, and trehalases, secreted by their midguts [3]. The enzymatic breakdown of intricate plant compounds into more manageable forms is vital for insects to obtain the essential nutrients required for their growth and development. In order to overcome the defensive obstacles posed by plant metabolites, insects have evolved mechanisms to adapt and optimize the activity and expression of their digestive enzymes [4,5].

Furthermore, the insect midgut hosts a diverse conjunto of antioxidative enzymes, including superoxide dismutase, catalase, and peroxidases [6,7]. These enzymes play a critical role in cellular homeostasis by scavenging reactive oxygen species (ROS) generated during digestion and other metabolic processes. While antioxidative enzymes can contribute to the detoxification of certain plant secondary metabolites, their primary function is to maintain cellular integrity and protect against general oxidative damage [8]. This enzymatic defense system is essential for the physiological resilience of insects, enabling them to thrive on diverse diets.

The resistance of insects to plant metabolites is influenced by the biochemical and physiological characteristics of their midguts. Insects can secrete a diverse range of digestive enzymes and defense enzymes, which aid in combating the plant’s defense mechanisms [9,10,11].

Several studies have investigated the impact of secondary metabolites on insect pests. For instance, research on Meliaceae plants examined how these metabolites influence digestive enzymes, affecting feeding behavior, digestion, and feeding efficiency in lepidopteran pests. The key enzymes studied included alpha-amylase, beta-glucosidase, lipase, and serine protease, as well as cysteine and aspartic proteinases. Additionally, the study explored the effects of these metabolites on the venom-producing enzymes of these insects, specifically examining enzymes such as general esterases (EST), glutathione S-transferases, and phosphatases [12]. A separate study examined the impact of secondary metabolites derived from the *Quercus robur* L. oak tree on *Tortrix viridana* L. revealing that the insects could effectively break down both primary and secondary metabolites, leading to improved food absorption efficiency [13]. Detoxifying enzymes and antioxidant enzymes in insects serve distinct but occasionally overlapping roles in managing toxic challenges. Detoxifying enzymes, such as cytochrome P450 monooxygenases (CYPs), glutathione S-transferases (GSTs), and carboxylesterases, primarily neutralize xenobiotics like pesticides and plant allelochemicals through metabolic processes that modify, conjugate, and excrete these toxins. In contrast, antioxidant enzymes, including superoxide dismutase (SOD), catalase (CAT), and glutathione peroxidases (GPX), counteract reactive oxygen species (ROS) generated during metabolic stress or environmental exposure, preventing oxidative damage to cellular components. While GSTs play a dual role in both detoxification and antioxidant defense, detoxifying enzymes are predominantly involved in xenobiotic metabolism, whereas antioxidant enzymes focus on ROS scavenging. Together, these enzyme systems enable insects to survive toxic challenges and maintain cellular homeostasis [7,14,15,16].

A different investigation assessed the effects of various plants, including purslane, chives, parsley, basil, dill, coriander, and mint, on the activities of amylolytic and proteolytic digestive enzymes in *Spodoptera littoralis* Boisduval, 1833. The results indicated that coriander exhibited the highest enzyme activity, while purslane showed the lowest, suggesting that purslane may be unsuitable for this insect due to the inhibitory effects of its metabolites on enzyme activities [17]. In another study examining different sugarcane cultivars and their effects on the pink stem stalk borer, *Sesamia cretica* Lederer,1857, cultivar CP73-21 demonstrated the greatest suitability for cultivation, while cultivar CP57-614 exhibited the lowest levels of amylolytic activity, likely due to distinct phytochemical metabolites present in the cultivars [18]. Research by Qi et al. (2022) [19] explored the population dynamics and variations in digestive enzymes (amylase, lipase, and trypsin) of the thrips, *Frankliniella occidentalis* Pergande, 1895, on various cultivars of *Rosa chinensis* Jacq., including Ruby, Love, Parade, Pink Peace, and Mohana. The findings indicated a direct correlation between the metabolites in different rose cultivars and the activity of thrips’ digestive enzymes, influencing population growth. Notably, the Ruby variety exhibited the highest amylase and lipase activity, facilitating rapid growth of the thrips, while the Pink Peace cultivar showed the highest trypsin activity. The Mohana variety, in contrast, had the lowest activity levels for all three digestive enzymes. Consequently, the researchers concluded that the population growth of *F. occidentalis* in relation to *R. chinensis* cultivars may be influenced by the activity of their digestive enzymes, which play an undeniable role in nutrient metabolism and insect growth. Furthermore, a study on the antioxidant enzymes of the whitefly *Bemisia tabaci* Gennadius, 1889 and their correlation with metabolites in cucumber plants (including Ghazeer, Melouky, and Zeen varieties) found that the Melouky variety, which had higher levels of secondary metabolites, led to increased activity of catalase and superoxide dismutase enzymes in this whitefly species [20].

Importantly, research indicates that an increase in plant metabolites does not always correlate with enhanced resistance against insect pests, nor are these metabolites consistently detrimental to insects. For example, a study examining the phenolic content of tobacco leaves in relation to the *Heliothis virescens* Fabricius, 1777, insect found that high concentrations of plant phenols did not significantly enhance the plant’s resistance to this pest. Instead, subsequent investigations revealed that the phenolic content in tobacco leaves improved the antioxidant capacity of the insect itself [21].

In conclusion, achieving a comprehensive understanding of the intricate interplay between plant metabolites, insect digestive enzymes, and antioxidative enzymes is important for unraveling the mechanisms that govern insect–plant interactions. By exploring the dynamic relationships among these factors, researchers can gain valuable insights into the strategies that insects use to circumvent plant defenses and the potential counter-adaptations that plants may evolve to deter herbivory. This knowledge can inform the development of innovative pest management strategies that specifically target these interactions, paving the way for more sustainable and effective approaches to crop protection. Figure 1 presents a general diagram illustrating the relationships between plants, insect pests, and the biotic and abiotic factors that influence both plant and insect biology.

## 2. Insect Enzymes

### 2.1. Insect Digestive Enzymes

Insects are formidable pests that pose significant challenges to agricultural productivity and food security. Traditional pest control methods often rely on the use of chemical pesticides, which can have detrimental effects on the environment and human health [22]. Therefore, there is a growing need for sustainable and environmentally friendly approaches to manage pest populations [23]. One promising avenue is the regulation of digestive enzyme activity in insects, which plays a central role in their ability to efficiently acquire nutrients from plant hosts [24].

Insect digestive enzymes are essential for the breakdown of complex macromolecules present in plant tissues into smaller and absorbable units. These enzymes are produced in specialized tissues within the insect’s digestive system, such as the midgut, and are tightly regulated to ensure efficient nutrient acquisition [25,26]. Recent studies have shed light on the intricate mechanisms underlying the regulation of digestive enzyme activity in insects, revealing the involvement of various factors, including gene expression, enzyme synthesis, and post-translational modifications [27,28,29].

Understanding the mechanisms by which insects regulate their digestive enzyme activity is of great importance in the context of plant–insect interactions. In response to insect herbivory, plants have evolved sophisticated defense mechanisms, including the production of protease inhibitors and other anti-nutritional factors that can interfere with the activity of insect digestive enzymes [30]. Conversely, insects have developed strategies to overcome these plant defenses and optimize their nutrient acquisition [29]. By unraveling the intricate interplay between insect digestive enzymes and plant defenses, researchers can gain insights into the co-evolutionary arms race between insects and their host plants.

Moreover, the understanding of the regulation of digestive enzyme activity in insects has significant applications in the field of pest management. Targeting digestive enzymes using inhibitors and genetic engineering techniques holds promise for the development of effective biopesticides [31,32]. By selectively inhibiting key enzymes involved in nutrient acquisition, pest insects can be controlled without the use of broad-spectrum chemical pesticides [33]. Additionally, the manipulation of plant metabolites to deter or inhibit specific digestive enzymes in pest insects offers the potential to develop pest-resistant crops, reducing the reliance on synthetic insecticides and promoting sustainable agriculture practices [8,24,34].

Despite the progress made in understanding the regulation of digestive enzyme activity in insects, there are several future directions that need to be explored. Research efforts should focus on identifying novel plant metabolites with enzyme-modulating properties and elucidating the underlying molecular mechanisms [29]. Moreover, the development of more specific and efficient enzyme inhibitors is essential to minimize off-target effects. Assessing the ecological impacts and long-term efficacy of enzyme-based control strategies in different agricultural systems and environments is also necessary [24]. Furthermore, the regulatory approval and public acceptance of genetically modified crops and biopesticides need to be addressed for successful implementation.

### 2.2. Insect Antioxidative Enzymes

Reactive oxygen species (ROS) are natural byproducts of metabolism and play important roles in various physiological processes. However, excessive ROS production can lead to oxidative stress, causing damage to cellular components and contributing to the development of various diseases [35,36,37]. To counteract the harmful effects of ROS, organisms have evolved antioxidant defense systems. Insects, as highly diverse and adaptable organisms, possess their own unique antioxidant mechanisms to maintain redox homeostasis [7].

In recent years, research on insect antioxidants has gained significant attention due to their potential implications in various aspects of insect biology, including development, reproduction, immune response, and response to environmental stressors [8]. Antioxidant enzymes, such as superoxide dismutase (SOD), catalase (CAT), and glutathione peroxidase (GPX), play key roles in scavenging ROS and maintaining the balance between oxidative stress and antioxidant defense [38,39]. Other non-enzymatic antioxidants, such as glutathione, ascorbic acid, and tocopherols, also contribute to the overall antioxidant capacity of insects [7,40]. Superoxide dismutase (SOD) is a critical metalloenzyme that catalyzes the dismutation of superoxide radicals (O_2_^−^) into hydrogen peroxide (H_2_O_2_) and molecular oxygen (O_2_), thereby playing a pivotal role in mitigating oxidative stress. This enzyme exists in multiple isoforms, including the copper-zinc variant (Cu/Zn-SOD), which is predominantly localized in the cytosol, and the manganese-containing form (Mn-SOD), which is primarily situated within the mitochondrial matrix. These distinct isoforms ensure the enzymatic neutralization of superoxide radicals across various cellular compartments, safeguarding cellular integrity and function [41]. Catalase (CAT) is a crucial antioxidant enzyme that facilitates the decomposition of hydrogen peroxide (H_2_O_2_) into water (H_2_O) and molecular oxygen (O_2_), thereby mitigating the potential for oxidative damage associated with the excessive accumulation of H_2_O_2_. This enzymatic reaction is essential for maintaining cellular redox homeostasis and protecting biomolecules from oxidative stress [42]. Glutathione peroxidase (GPx) is a vital selenoenzyme that plays a central role in cellular defense against oxidative stress by catalyzing the reduction of hydrogen peroxide (H_2_O_2_) and organic hydroperoxides to water (H_2_O) and their corresponding alcohols. This reaction utilizes reduced glutathione (GSH) as an electron donor, leading to the formation of oxidized glutathione disulfide (GSSG). The catalytic mechanism involves the oxidation of the enzyme’s active-site selenol group (-SeH) by peroxides to form a selenenic acid intermediate (-SeOH), which is subsequently reduced through two sequential reactions with GSH. This process not only neutralizes harmful peroxides but also maintains redox homeostasis within cells, with GSSG being recycled back to GSH by glutathione reductase in an NADPH-dependent manner [43]. Glutathione S-transferases (GSTs) in insects are multifunctional enzymes primarily recognized for their role in detoxification, facilitating the conjugation of glutathione to a wide range of electrophilic substrates, including xenobiotics and endogenous compounds. Beyond detoxification, GSTs are integral to cellular defense mechanisms against oxidative stress by neutralizing reactive oxygen species (ROS) and their by-products, such as lipid peroxidation derivatives. This enzymatic activity not only mitigates oxidative damage but also contributes to the regulation of redox homeostasis, underscoring the critical role of GSTs in insect physiology and stress response [44]. Peroxiredoxins (Prxs) are thiol-based peroxidase enzymes that play a pivotal role in cellular defense against oxidative stress by reducing hydrogen peroxide and organic hydroperoxides through cysteine-dependent catalytic mechanisms. These enzymes not only mitigate oxidative damage but also regulate redox signaling pathways, influencing processes such as gene expression, enzyme activity, and cellular adaptation to stress. By maintaining redox homeostasis, Prxs contribute to the preservation of cellular integrity and functionality under conditions of oxidative stress [45,46].

Insect antioxidants have been shown to have diverse functions beyond redox regulation. For example, they have been implicated in the modulation of insect lifespan, reproduction, and response to environmental stressors such as temperature extremes, insecticides, and pathogens [7,8,40]. Understanding the mechanisms underlying the regulation of insect antioxidant systems can provide valuable insights into their adaptation to changing environmental conditions and their interactions with other organisms.

Furthermore, the study of insect antioxidants has practical implications in the context of pest management and disease control. Insects, including agricultural pests and disease vectors, often face oxidative stress as a result of exposure to pesticides, pathogens, and environmental pollutants [8]. Recent research has further shown that the increase in plant antioxidants influences the life table and biological stages of pest insects [47]. In some instances, this effect leads to a shorter developmental period for the pest insects and a reduction in their reproduction and survival rates [23]. Exploiting the vulnerabilities in their antioxidant defense systems can offer novel strategies for developing targeted insecticides and disease control measures. Additionally, the manipulation of antioxidant pathways in pest insects may provide avenues for enhancing the efficacy of existing control methods or developing new approaches that exploit their oxidative stress vulnerabilities [48,49,50].

## 3. Plant Metabolites and Their Impact on Insect Enzymes

### 3.1. Plant Metabolites and Their Impact on Digestive Enzymes

Plant secondary metabolites exhibit a dual nature, capable of both benefiting and harming insects and humans. These compounds play crucial roles in plant defense mechanisms against herbivores, while also serving as valuable resources for pharmaceutical and nutraceutical industries [51]. Dietary changes have been shown to induce significant shifts in insect metabolism, affecting enzymatic profiles, protein content, fatty acid composition, and associated enzymes [34]. During feeding, insects encounter a wide array of plant metabolites that can profoundly influence their digestive enzyme activity. Secondary metabolites, such as alkaloids, phenolics, and terpenoids, are commonly found in plants and often serve as defense compounds against herbivory [8,52]. These secondary metabolites can have both positive and negative effects on insect digestive enzymes. For instance, certain alkaloids have been shown to inhibit the activity of insect amylases and proteases, impairing carbohydrate and protein digestion [53,54]. On the other hand, some phenolic compounds, such as tannins, can enhance the activity of insect digestive enzymes, potentially aiding in the breakdown of dietary components [6,55,56]. Allelochemicals, which are compounds released by plants to influence the growth and behavior of other organisms, can also impact insect digestive enzymes [57]. For example, certain allelochemicals have been found to inhibit the activity of insect total protease, glucosidase, and lipases in *Lymantria dispar* L. [58]. Understanding the mechanisms by which plant metabolites modulate digestive enzyme activity is necessary for comprehending the intricate dynamics of insect–plant interactions. One of the important consequences of pest control tactics arises from the regulation of digestive enzyme activity in insects. Many studies have been conducted on the digestive enzymes of insects, including enzyme characteristics at different stages of the insect [59], the impact of various laboratory compounds [60], the effects of temperature [61], and enzyme inhibitors extracted from plants [62]. Gaining knowledge of the regulation of these enzymes can help with the development of efficient control strategies. Using enzyme inhibitors to target the digestive enzymes directly is one strategy [53,62,63,64]. For example, Askari et al. [62] reported that plant seed extracts inhibited the digestive protease of *Helicoverpa armigera* Hübner, 1808. In another study, Ashouri and Farshbaf Pourabad [28] demonstrated that the white bean and rapeseed protein extracts altered the expression of the digestive α-amylase gene in the Colorado potato beetle, *Leptinotarsa decemlineata* (Say, 1824). These inhibitors can disrupt the digestion process in pest insects, leading to reduced nutrient absorption and ultimately, decreased fitness and survival. Another strategy is to manipulate the expression of digestive enzymes through genetic engineering. By introducing genes that encode digestive enzyme inhibitors or altering the expression of specific enzymes, it is possible to impair the ability of pests to efficiently digest their food [65,66]. Genetic modification of plants to inhibit insect digestion utilizes plant protease inhibitors (PPIs), which disrupt insect gut proteases, impairing digestion and reducing fitness. However, insects counteract PPIs through adaptive gene expression and protease modification. Advanced biotechnological tools like CRISPR-Cas and RNA interference enhance PI gene expression or suppress insect protease genes, improving pest resistance. While transgenic plants overexpressing PPIs show potential, their effects on non-target organisms and the role of gut microbiota in protease regulation must be considered for sustainable pest management [67]. Since the discovery of gene expression silencing via double-stranded RNA (dsRNA) over two decades ago, RNA interference (RNAi) has emerged as a promising tool for agricultural pest management. For example, the first practical application involved transgenic maize expressing dsRNA targeting vacuolar ATPase, reducing Diabrotica virgifera virgifera (western corn rootworm) feeding [68]. Various dsRNA delivery methods, including direct injection and virus-induced gene silencing (VIGS), have since been explored for pest control [69]. These are brief examples of genetic strategies employed to manipulate insect feeding, aiming to develop highly pest-resistant plants. Undoubtedly, other approaches also exist; however, a detailed discussion of them falls beyond the scope of this work.

Additionally, understanding the interaction between plant metabolites and digestive enzymes can also inform the development of pest-resistant crops. Overall, the regulation of digestive enzymes in insects provides a valuable target for pest management strategies. Table 1 summarizes selected studies investigating the effects of plant metabolites on insect digestive enzymes.

The investigations carried out on the insecticidal and nutritional inhibitory properties of azadirachtin against *Drosophila melanogaster* revealed that various species belonging to the Meliaceae genus, such as *Azadirecta indica*, exhibit inhibitory effects on alpha-amylase, chitinase, and protease enzymes, while concurrently enhancing lipase activity. Moreover, it was observed that this plant species can diminish larval feeding and exert a noteworthy influence on food digestion capabilities [81]. To examine the impact of plant secondary metabolites on insects, a research study was conducted to assess the effects of seven monoterpenes, two phenylpropenes, and two sesquiterpenes on the second instar larvae of *Spodoptera littoralis*. The findings revealed significant anti-nutritional activity, growth inhibition, and insecticidal properties associated with these compounds. Additionally, these compounds demonstrated potent inhibition of digestive alpha-amylase and total protease enzymes, further contributing to their detrimental effects on the larvae [89]. Earlier research on beta-caryophyllene primarily focused on its toxic properties as an insecticide. However, there has been increasing interest in exploring its anti-nutritional characteristics. An investigation was carried out to examine the impact of the *Ocimum kilimandscharicum* Gürke plant on *Helicoverpa armigera*. The study revealed that following an attack by the cotton bollworm on this plant, there was a notable increase in the levels of beta-caryophyllene, camphor, and limonene. These chemical compounds were found to contribute to weight loss in larvae, an elevation in larval mortality rates, and significant alterations in digestive enzymes such as amylase, protease, and lipase. These findings highlight the substantial effects of *O. kilimandscharicum* on various physiological aspects of *H. armigera* larvae in response to pest infestation [90]. Consequently, Mahajan et al. [91] conducted a study to investigate the impact of this sesquiterpene on the feeding physiology of *Spodoptera litura* Fabricius larvae. The results revealed that beta-caryophyllene exerted a detrimental effect on various parameters related to feeding efficiency, including the conversion efficiency of ingested food, the conversion efficiency of digested food, and approximate digestibility. These findings contribute to our understanding of the negative influence of beta-caryophyllene on the feeding biology of *S. litura* larvae. A study was conducted to examine the impact of quercetin on the enzymatic activities of the diamondback butterfly (*Plutella xylostella* (L.)), with a focus on alpha-amylase, protease, and lipase. The findings revealed that quercetin exhibited significant inhibitory effects on the enzymatic activities of the diamondback butterfly. These results suggest that quercetin has the potential to disrupt the digestive processes and metabolic functions of the butterfly species under investigation [92]. An intriguing investigation examining the impact of plant metabolites on the activity of the cotton bollworm (*Helicoverpa armigera*) revealed that various plant flavonoids, including quercetin, cinnamic acid, caffeic acid, chlorogenic acid, catechin, trihydroxy flavone, gentisic acid, ferulic acid, protocatechuic acid, and umbelliferone, as well as lectin and phenyl β-d-glucoside, exhibited adverse and inhibitory effects on insect digestive enzymes such as total serine protease and trypsin, as well as detoxifying enzymes like esterases and glutathione S-transferase [93]. Figure 2 provides an overview of plant chemical defenses and illustrates how insect pests interact with and respond to these defensive compounds.

### 3.2. Plant Metabolites and Their Impact on Insect Antioxidative Enzymes

The impact of plant secondary metabolites on the antioxidant system of insects has been extensively investigated in select plant species and their associated pests. However, the number of studies conducted in this area remains relatively limited compared to research focusing on the influence of secondary metabolites on insect digestive enzymes. Nevertheless, noteworthy research has been conducted, particularly in relation to the activation of antioxidant enzymes in insects [94]. The intersection of plant secondary metabolites and insect antioxidative enzymes is a central topic in ecological and evolutionary research, with these plant chemicals possessing a diverse range of impacts on insects, both beneficial and detrimental [95]. Numerous studies have demonstrated that phytophagous insects would eventually produce a certain adaptability to the defense of host plants after ingesting the secondary metabolites of those plants. These mechanisms would improve the insects’ detoxification and metabolic capacity toward secondary metabolites [96,97].

The production of (ROS) poses a significant threat to cellular functioning, arising both from natural metabolic byproducts and as a consequence of diverse stressors. Insects have adapted sophisticated mechanisms to manage ROS and maintain cellular homeostasis.

Secondary metabolites produced by plants can interact with the antioxidative enzymes of insects in several distinct ways. Some of these metabolites may serve as prooxidants, leading to the induction of oxidative stress in insect organisms by altering the delicate balance between the generation of (ROS) and their subsequent detoxification. Such an imbalance can cause significant oxidative harm to various cellular components, including lipids, proteins, and genetic material like DNA [98,99]. Conversely, specific secondary metabolites can function as antioxidants by either directly neutralizing (ROS) or indirectly influencing the activity of antioxidative enzymes. For instance, certain phenolic compounds and flavonoids present in plants have been demonstrated to enhance the activities of superoxide dismutase (SOD), catalase (CAT), and glutathione peroxidase (GPX) in insects, thereby bolstering their antioxidative defense mechanisms [100].

The effects of plant secondary metabolites on insect antioxidative enzymes can have significant ecological and evolutionary implications. Insects that are exposed to high levels of secondary metabolites might experience increased oxidative stress, which can affect their fitness, behavior, and interactions with their environment [101,102]. Furthermore, the ability of insects to tolerate or detoxify plant secondary metabolites can influence their host range, feeding preferences, and adaptation to new plant species [103]. In Jiang et al.’s study [104] on *Hyphantria cunea* Drury larvae, it was observed that the inclusion of cinnamic acid in the diet did not result in any harmful effects and did not lead to an increase in oxidative stress parameters such as malondialdehyde and hydrogen peroxide. However, cinnamic acid did induce an enhancement in the insect’s antioxidant system, including enzymatic components such as superoxide dismutase and peroxidase, as well as non-enzymatic components like glutathione and ascorbic acid. Additionally, it raised the concentration of detoxifying enzymes such as carboxylesterase. Based on these results, one may deduce that plant metabolites, despite their lack of detrimental impact on insect oxidative processes, can still elicit a stimulatory response from the insect’s defense mechanisms. The absence of harmful effects may be attributed to either the relatively low concentration of the plant metabolite or the insect’s efficient enzymatic capacity to effectively metabolize and neutralize these compounds.

Understanding the relationship between insect antioxidative enzymes and plant secondary metabolites is essential for unraveling the mechanisms underlying insect–plant interactions [8].

## 4. Antioxidative Enzymes and Oxidative Stress

Insects face various environmental stressors that can induce oxidative stress, leading to the production of excessive (ROS). Insects mitigate oxidative damage through a complex antioxidative enzyme network, where superoxide dismutase (SOD) converts superoxide radicals into hydrogen peroxide (H_2_O_2_), [93] which is then detoxified into water and oxygen by catalase (CAT) and glutathione peroxidase (GPX) [8].

The expression and activity levels of antioxidative enzymes in insects can be modulated in response to various stressors. For instance, exposure to pesticides [94], heavy metals [8], or pathogens [95] can induce an upregulation of antioxidative enzymes as a defense mechanism against oxidative stress. Conversely, environmental factors such as temperature extremes [96], UV radiation [97], and nutrient availability [8] can influence the expression and activity of these enzymes. The regulation of antioxidative enzymes in insects is complex and involves an intricate interplay between transcription factors, signaling pathways, and post-translational modifications [98].

Furthermore, the role of oxidative stress and antioxidative enzymes extends beyond redox homeostasis. Insects have evolved mechanisms to utilize oxidative stress as signaling molecules, modulating various physiological processes including development, reproduction, and immune response [99,100]. For example, studies have shown that ROS can act as secondary messengers in insect immune signaling pathways, regulating the expression of antimicrobial peptides and immune-related genes in black soldier flies *Hermetia illucens* L. [101]. Additionally, oxidative stress has been linked to insect lifespan, with elevated levels of ROS resulting in accelerated aging and reduced longevity [102,103,104]. Understanding the intricate connections between oxidative stress, antioxidative enzymes, and physiological processes in insects can provide valuable insights into their adaptation and survival strategies. The interplay between digestive and antioxidative enzymes in insects represents a critical facet of their adaptation to plant defense mechanisms, particularly in the context of reactive oxygen species (ROS) generated during digestion. ROS, including superoxide radicals (O_2_^−^) and hydrogen peroxide (H_2_O_2_), are produced both exogenously from plant allelochemicals and endogenously through metabolic processes. This dual origin underscores the complex redox dynamics essential for insect survival and their evolutionary responses to phytochemical stressors [7,8,105]. These ROS can damage the active sites of proteases, impair digestive enzyme functionality, and reduce nutrient assimilation. For instance, ROS-induced oxidation of cysteine residues in proteases can lead to enzyme inactivation, while lipid peroxidation may disrupt membrane-bound digestive enzymes. Such oxidative modifications highlight the intricate balance between ROS generation and antioxidative defense mechanisms in maintaining enzymatic efficiency and insect nutritional physiology [8,106,107]. Research on the *Mayetiola destructor* Say (Hessian fly) reveals that larvae in resistant wheat exhibit elevated expression of antioxidant genes (MDESPHGPX-1, MDESCAT-2) to alleviate ROS-induced stress, indicating a direct connection between oxidative stress and digestive efficiency [108]. Detoxification enzymes like GSTs and β-glucosidases enhance resistance by degrading plant toxins and mitigating ROS, highlighting their role in insect–plant interactions and adaptive evolution.

Despite significant advancements in our understanding of antioxidative enzymes and oxidative stress in insects, several research directions warrant further exploration. Firstly, investigating the role of non-enzymatic antioxidants, such as glutathione, ascorbic acid, and tocopherols, in insect redox homeostasis and stress response would provide a more comprehensive understanding of their antioxidant defense mechanisms [105,106]. Secondly, unraveling the molecular mechanisms underlying the regulation of antioxidative enzymes in response to different stressors and environmental conditions will shed light on the plasticity and adaptability of insect antioxidative systems as mentioned before (like gene expression, post-translational regulation, etc.). Moreover, exploring the interplay between antioxidative enzymes and other cellular processes, such as energy metabolism and signal transduction pathways, will provide a more integrated view of insect physiology and stress response. Figure 3 presents a conceptual overview of oxidative stress and the physiological mechanisms employed by both plants and insects to mitigate its effects.

These studies have assessed the response of antioxidant enzymes to both biotic and abiotic stressors, including heavy metals, temperature fluctuations, and biocontrol agents such as fungi, nematodes, and bacteria. The list of these investigations are summarized in Table 2.

## 5. Interplay Between Digestive and Antioxidative Enzymes

Insects possess a unique interplay between digestive enzymes and antioxidative enzymes, which is essential for their survival and adaptation to different dietary sources. Digestive enzymes, such as proteases, lipases, and carbohydrases, are responsible for the breakdown and utilization of ingested food components [29]. However, the process of digestion can also generate oxidative stress due to the production of (ROS) during metabolic reactions, especially in feeding plants that have defensive metabolites. To counteract this oxidative stress and maintain redox balance, insects have evolved a coordinated system involving both digestive and antioxidative enzymes. The presence of high levels of antioxidant enzymes (SOD and GPX) in the foregut and hindgut compartments suggests that they play a role in protecting the insect gut and its microbiota from oxidative damage. These compartments also have cuticular layers that provide additional protection against physical and chemical harm to the cells’ surface [138]. In contrast, the midgut, which lacks cuticular layers, exhibits high levels of oxidants along with low transcript levels and enzymatic activities of SOD and GPX. This indicates that the cells in the midgut are more vulnerable to oxidative damage [139].

Recent studies have shed light on the intricate connections between digestive and antioxidative enzymes in insects. For instance, it has been observed that the activity of digestive enzymes can be influenced by the redox status of the insect gut. Oxidative stress can impair the activity of digestive enzymes, leading to reduced nutrient absorption and compromised digestive efficiency [140]. Conversely, the activity of antioxidative enzymes can be regulated by the availability of dietary antioxidants and the composition of ingested food. Certain dietary components, such as polyphenols and flavonoids, have been shown to enhance the activity of antioxidative enzymes in insects, thereby mitigating oxidative stress [141].

Moreover, the interplay between digestive and antioxidative enzymes extends beyond their individual functions. It has been proposed that digestive enzymes can also possess antioxidant properties, providing an additional layer of protection against oxidative stress. For example, some studies have reported the antioxidant activity of digestive enzymes, such as trypsin and chymotrypsin [142]. These enzymes can scavenge ROS directly, thereby reducing the oxidative burden within the insect gut. Additionally, certain digestive enzymes have been found to enhance the absorption and utilization of dietary antioxidants, further supporting the antioxidative defense system in insects [29].

The interplay between antioxidant and digestive enzymes is thought to be important for a number of reasons. First, antioxidant enzymes can protect digestive enzymes from damage by ROS [143]. Second, digestive enzymes can generate ROS, which can then be scavenged by antioxidant enzymes [144]. Third, antioxidant enzymes can help to protect the gut from damage by ROS produced by bacteria or other microorganisms [145]. Research on the interplay between antioxidant and digestive enzymes is still in its early stages, but it has already revealed some important insights. For example, Khochapong et al. [146] have shown that antioxidant enzymes can enhance the activity of digestive enzymes in insects. In addition, studies have shown that digestive enzymes can increase the production of ROS in insects [144]. These findings suggest that the interplay between antioxidant and digestive enzymes is a complex and dynamic process that is likely to play an important role in insect health and fitness. Increasing the activity of antioxidant enzymes can protect beneficial insects, like pollinators, from the damaging effects of ROS [8]. Similarly, reducing the production of ROS by digestive enzymes can improve food digestibility for insects [135]. Additionally, targeting the antioxidative defense system in pest insects can offer a novel approach to pest control.

## 6. Regulation of Enzyme Activity in Response to Plant Metabolites

The regulation of enzyme activity in response to plant metabolites is an essential aspect of insect–plant interactions. Plant metabolites can act as inducers or inhibitors of digestive enzymes in insects, thereby influencing their ability to efficiently utilize plant resources. For instance, certain secondary metabolites such as allelochemicals can induce the synthesis of specific enzymes involved in detoxification or digestion processes [22]. On the other hand, some plant metabolites can inhibit key digestive enzymes, leading to reduced nutrient acquisition by insects [62]. The regulation of enzyme activity in response to plant metabolites is often mediated through signaling pathways, gene expression changes, and post-translational modifications [27,28]. Understanding the mechanisms underlying this regulation is essential for deciphering the complex interactions between insects and their host plants. Furthermore, the identification and characterization of specific plant metabolites that can modulate enzyme activity can have significant implications for the development of novel pest management strategies. Studies have demonstrated that certain plant metabolites, such as flavonoids and phenolics, can induce the upregulation of antioxidant enzymes in insects [4,8].

While the regulation of digestive and antioxidative enzyme activity in insects holds great promise for pest management, there are several limitations and challenges that need to be addressed. Firstly, the specificity of enzyme inhibitors needs to be improved to minimize off-target effects on non-target organisms [22]. Additionally, the development of resistance by pest insects to enzyme inhibitors is a potential concern, as seen with other pest control methods. An extensive investigation has revealed the intricate mechanisms employed by insects to circumvent the presence of plant-derived enzyme inhibitors, thereby ensuring the continued functionality of their digestive enzymes or developing adaptations to cope with these inhibitors over a prolonged period [147]. Studies indicate that certain insects can efficiently bypass plant metabolite barriers, reducing their toxicity or rendering them non-toxic. For example, the larvae of *Pieris rapae* L. (cabbage white) utilize a specialized gut protein known as a nitrile-specific protein (NSP) to modify the glucosinolate breakdown pathway, leading to the production of less toxic nitriles instead of harmful isothiocyanates [148]. In plants, glucosinolates are typically hydrolyzed by the enzyme myrosinase, resulting in the formation of toxic isothiocyanates. However, in the larval gut, nitrile-specific protein (NSP) redirects this reaction toward the production of nitriles, such as 4-hydroxyphenylacetonitrile, which exhibit significantly lower toxicity [149]. In other insects, the detoxification of glucosinolates occurs through various mechanisms. For example, the diamondback moth *Plutella xylostella* utilizes the enzyme sulfatase to inactivate glucosinolates. This enzyme removes the sulfate group from glucosinolates, converting them into non-toxic compounds that no longer pose a defense threat to the insect [150]. Some insects eliminate toxic compounds through their feces or urine. For instance, *Manduca sexta* larvae excrete toxic compounds via their excretory system [151]. In the context of insect resistance to plant secondary metabolites, cytochrome P450 monooxygenases (CYPs) play a pivotal role in the detoxification of plant-derived xenobiotics, including alkaloids, terpenoids, and other defensive compounds. These enzymes catalyze oxidative reactions that enhance the solubility and excretion of toxic substances, thereby mitigating their detrimental effects. *Helicoverpa zea* Boddie utilizes a specific subset of CYP enzymes to metabolize plant defense chemicals, contributing to its ability to overcome chemical barriers and successfully exploit a wide range of host plants [152]. As part of their adaptive resistance to plant secondary metabolites, some insects avoid the toxic effects of these compounds by either preventing their absorption or selectively sequestering them in specific tissues. This strategy minimizes toxicity while allowing the insect to feed on chemically defended plants. *Zonocerus variegatus* L., for example, accumulates plant-derived toxic compounds in its fat bodies, effectively isolating them from physiologically sensitive systems. This sequestration not only reduces toxicity but may also serve as a defense mechanism against predators [153]. Also, the silkworm *Bombyx mori* exhibits specialized transcriptional and physiological adaptations to mulberry-derived sugar-mimic alkaloids, 1-deoxynojirimycin (1-DNJ), and 1,4-dideoxy-1,4-imino-D-arabinitol (D-AB1). Global transcriptional analysis revealed that both alkaloids trigger distinct but overlapping gene expression trajectories, modulating detoxification networks, digestive processes, and carbohydrate/lipid metabolism. While 1-DNJ elicited resilience through upregulated detoxification pathways, D-AB1 predominantly downregulated genes involved in carbohydrate catabolism and glycan biosynthesis, disrupting metabolic homeostasis more severely [154]. Parthenogenetic weevils (*Naupactus cervinus* Boheman and *N. leucoloma* Boh. in Schoenh) rely on transcriptional plasticity to adapt to diverse host plants, despite low genetic variation. Comparative transcriptomics revealed host-specific gene expression patterns, particularly in detoxification, immune defense, and host detection pathways. Taxing hosts, such as legumes, elicited complex and intense responses, yet shared differentially expressed genes with other stressors like novel hosts or cultivation conditions, suggesting a conserved gene expression regime for stress adaptation. Notably, elements of these responses were heritable, highlighting the role of epigenetic regulation in colonization success [155]. The role of glucosidase II (GII) in mulberry-specialist insects was investigated in the context of plant-herbivore co-evolution. GIIα expression was significantly induced by mulberry leaf extract and 1-deoxynojirimycin (1-DNJ), while GIIβ expression remained unchanged. Positive selection was detected in GIIα, indicating asymmetrical evolutionary pressures, with key mutations near 1-DNJ-binding regions and the C-terminal domain, potentially influencing enzymatic function. These findings provide insights into the molecular adaptations underlying insect responses to plant defensive compounds [156]. Studies have revealed that Lepidopterans have evolved genetic mechanisms to tolerate plant defensive compounds, though the role of gut microbiota in detoxification remains unclear. 1-deoxynojirimycin (DNJ), a mulberry-derived secondary metabolite, inhibited the growth of various lepidopteran species, except for the specialist *Bombyx mori*. DNJ exposure altered the gut microbiota of early-instar silkworms, with certain bacterial species responding positively. A gut-derived strain, Pseudomonas fulva ZJU1, was identified as capable of degrading DNJ and enhancing DNJ resistance and growth in nonspecialist hosts. Genomic and transcriptomic analyses revealed ilvB as a key gene in DNJ metabolism, with CRISPR-Cas9 mutagenesis confirming its role. The loss of DNJ resistance in ilvB-deficient bacteria demonstrated a direct link between gut microbiota and detoxification in Lepidoptera. These findings provide new insights into microbial contributions to insect adaptation against plant chemical defenses [157]. Understanding the mechanisms underlying resistance and implementing strategies to mitigate its development will be critical. Furthermore, the complex nature of insect–plant interactions makes it challenging to predict the effectiveness of enzyme-based control strategies in different agricultural systems and environments [30]. More research is needed to assess the ecological impacts and long-term efficacy of these approaches. Finally, the regulatory approval and public acceptance of genetically modified crops and biopesticides pose additional challenges [158]. Addressing these limitations and challenges will be essential for the successful implementation of insect enzyme regulation strategies in pest management practices.

## 7. Applications and Future Directions

Future research in this field should focus on elucidating the specific mechanisms underlying the interactions between plant metabolites and insect enzymes. This includes identifying the key metabolites responsible for modulating enzyme activity, understanding the signaling pathways involved, and investigating the ecological implications of these interactions in natural ecosystems.

The discovery and characterization of new plant metabolites with potential use in pest management represents a promising direction for future research [24]. Using a variety of plant species as a filter, scientists can find novel bioactive substances that either interfere with insect antioxidant defense systems or specifically target the digesting enzymes of insects. In comparison to conventional chemical pesticides, this may result in the creation of innovative biopesticides or insect resistance boosters that have less of an adverse effect on the environment [24].

Furthermore, knowledge of the co-evolutionary processes between plants and insects might shed light on the competition between insect adaptations and plant defenses [159,160]. The mechanisms behind insect resistance or tolerance to plant defenses can be clarified by looking at the genetic and molecular basis of insect adaptations to plant metabolites [161]. This information may be used to create novel approaches to managing pests, such as using genetic engineering to strengthen plant defenses or tampering with insect digestive enzyme pathways.

Additionally, investigating the possibility of employing plant metabolites to generate systemic acquired resistance (SAR) in plants may have important ramifications for environmentally friendly pest control techniques. When a plant is exposed to certain substances, its defensive mechanism, known as SAR, is triggered across the plant, increasing its resistance to a variety of pests and diseases [162]. For instance, an investigation focused on the disruption of the phenylpropanoid pathway, a compound known as piperonylic acid (PA), which acts as an inhibitor of cinnamic acid 4-hydroxylase (C4H), was employed. The study revealed that the application of this inhibitor in plants resulted in a notable enhancement of systemic resistance against pests and diseases, without adversely affecting plant growth [163]. Researchers can boost plant resistance to insect pests by developing ecologically friendly methods by discovering plant metabolites that might cause SAR. Further research is needed to investigate the ecological consequences of interactions between plant metabolites and insect enzymes. Understanding how these interactions determine insect behavior, population dynamics, and community structure will be invaluable for understanding the functioning of natural ecosystems. Using genetic engineering and plant breeding techniques, researchers will be able to modify plant metabolites and predict their effects on insect herbivory and ecosystem dynamics.

The next stages of the research will focus on gaining a clearer understanding of the ecological and evolutionary implications of the interactions between plant metabolites and insect enzymes, looking for other plant metabolites that could be used for pest control, studying insect adaptations to the plant defense mechanisms, and studying the pests and beneficial insects such as predators and parasites. These research activities will aid in formulating strategies for pest control that are eco-friendly and help grasp the intricate relationship between plants and insects. However, the extent of research conducted in this particular domain remains limited, necessitating heightened attention within the field of agriculture. Thorough investigations should be undertaken to examine the impact of fertilization on plants and the subsequent alterations in both enzymatic and non-enzymatic plant metabolites. Existing studies have demonstrated that fertilization, encompassing micro and macro elements, has the potential to modify the abundance of plant metabolites, thereby exerting consequential effects on pest insects. These effects may further extend to tertiary trophic levels within the food chain.

One ethical suggestion stemming from the aforementioned text is to prioritize the safety and well-being of both humans and the environment when developing and implementing pest control strategies utilizing plant metabolites. While the potential benefits of using biopesticides or insect resistance boosters derived from plant metabolites are promising, it is essential to thoroughly assess the potential risks associated with their use. Rigorous testing should be conducted to evaluate the potential toxicity and unintended effects on non-target organisms, including beneficial insects, wildlife, and ecosystems. Additionally, transparency and public engagement should be prioritized to ensure that stakeholders have access to accurate information and can participate in decision-making processes regarding the adoption of these new technologies. By adhering to these ethical considerations, researchers and policymakers can ensure that the development and implementation of plant metabolite-based pest control strategies are conducted responsibly and sustainably.

## 8. Conclusions

Ultimately, the relationship between plant metabolites and insect antioxidative and digestive enzymes is very important in understanding the complexities of the interactions between insects and plants. Plants are self-defending by producing secondary metabolites that can shield them from the damaging effects posed by herbivorous insects. These metabolites can function as strong antioxidants, impairing the action of insect antioxidative enzymes and thwarting oxidative stress. Also, the action of insect digestive enzymes can directly be impacted by plant metabolites through inhibition of their activity or interfering with their binding sites. Their reduced ability to efficiently digest the plant tissues can impact their survival and fitness. On the other hand, insects have developed adaptive strategies to counter these defense mechanisms provided by plants. Insects also possess the ability to produce other protective substances that can eradicate these toxic substances along with their exotic organs. In addition, even insects are capable of synthesizing their antioxidative enzymes to neutralize the effects of plant metabolites. As a whole, the plant metabolites and insect antioxidative and digestive enzymes work together in a very active and responsive duel, in which both organisms ceaselessly change and improve to get the upper hand. For practical pest management and for understanding the intricate network of relationships between plants and insects in nature, it is critical to understand the peculiarities of this interplay.

## Figures and Tables

**Figure 1 insects-16-00392-f001:**
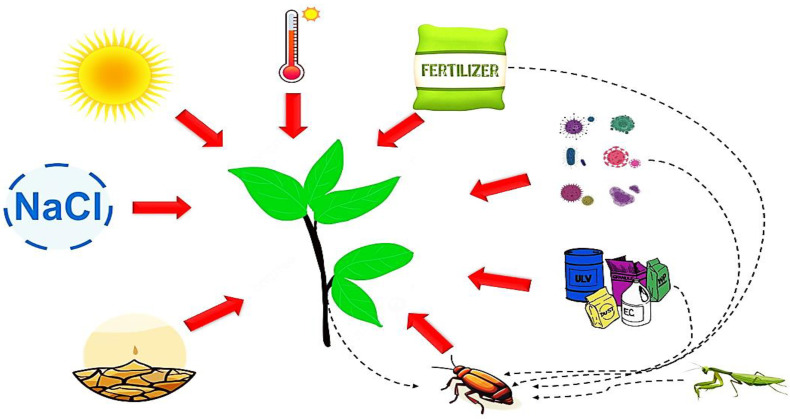
Interactive effects of environmental and anthropogenic factors on plant–insect dynamics, including the influence of abiotic stresses (salinity, heat, drought, light), biotic stresses (pathogens, pests), and natural enemies (parasitoids, predators). The arrows illustrate the factors influencing the plant. To protect the plant from biotic stressors such as pests and diseases, various measures including pesticides, fertilizers, and biological control agents are applied, all of which can also impact the plant. This figure provides an overview of the interactions between biotic and abiotic environmental factors affecting the plant, highlighting the complex relationships among these elements. The dotted lines indicate that factors such as fertilizers, insecticides, and biological pest control agents not only have direct effects on the plant but also influence the pest insect directly, and may indirectly impact the plant through their effects on the insect.

**Figure 2 insects-16-00392-f002:**
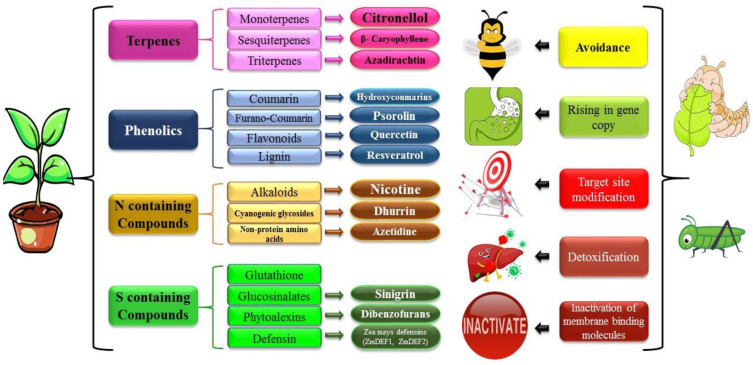
The co-evolutionary dynamics between plant chemical defenses and insect counter-adaptations. Plants produce a diverse array of secondary metabolites, including terpenes, phenolics, nitrogen-containing compounds, and sulfur-containing compounds, to deter herbivorous insects. In response, insects have evolved multiple strategies to overcome these defenses, such as avoidance behavior, gene duplication leading to increased detoxification capacity, target site modification to prevent toxin binding, enzymatic detoxification of harmful compounds, and inactivation of membrane-binding molecules. This intricate interplay drives the evolutionary arms race between plants and herbivorous insects, shaping ecological interactions and influencing species adaptation over time.

**Figure 3 insects-16-00392-f003:**
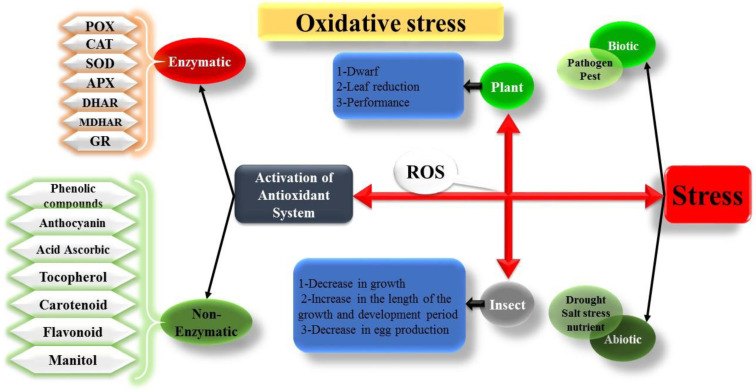
Interactive roles of reactive oxygen species (ROS) in plant and insect stress responses. Biotic and abiotic stressors induce ROS generation in both organisms. To counteract oxidative damage, plants and insects have evolved intricate antioxidant defense systems comprising enzymatic (e.g., SOD, CAT, POX) and non-enzymatic (e.g., flavonoids, ascorbic acid, carotenoids) components. The delicate balance between ROS production and antioxidant capacity determines the organism’s capacity to withstand stress and maintain cellular homeostasis.

**Table 1 insects-16-00392-t001:** Summary of research on insect digestive enzyme activity in response to various plants.

Researcher	Plant	Plant Utilization Form	Insect	Insect Stage	Investigated Enzymes
Mehrabadi et al. [70]	*Punica granatum* L. (Punicaceae) *Rheum officinale* Baill (Polygonaceae) *Rhus coriaria* L. (Anacardiaceae) *Artemisia sieberi* B. (Compositae) *Peganum harmala* L. (Nitrariaceae) *Datura stramonium* L. (Solanaceae) *Thymus vulgaris* L. (Lamiaceae)	Extract	*Callosobruchus maculatus* F. (Coleoptera: Bruchidae) *Rhyzopertha dominica* F. (Coleoptera: Bostrichidae) *Sitophilus granarius* L. (Coleoptera: Curculionidae) *Trogoderma granarium* E. (Coleoptera: Dermestidae).	Last larval instar	α-amylase
Sarate et al. [71]	Chickpea, pigeonpea Tomato, okra Rose, marigold Sorghum, maize	Herbal diet	*Helicoverpa armigera*	Larve pupa	Amylases Proteases Lipases
Napoleão et al. [72]	*Myracrodruon urundeuva* M.Allemão	Leaf extract and lectin	*Sitophilus zeamais* Motschulsky	adults	Protease Trypsin-Like Acid Phosphatase Amylase
Senthil-Nathan [12]	Meliaceae	Secondary metabolites	Pyralidae, Nuctoidae	(a review)	α-Amylases α and β-glucosidases Lipases Proteases Serine, Cysteine, And Aspartic Proteinases General Esterases (EST) Glutathione S-Transferase (GST) Phosphatases Alkaline Phosphatase Acid Phosphatase
Dastranj et al. [73]	wheat cultivars	Seed extract	*Tenebrio molitor* L.	All larval stages Adult	α-amylases Proteases
Sami [74]	*Azadirachta indica* A.Juss	Azadirachtin and Saponin	*Tribolium castaneum* Herbst *Aulacophora foveicollis* Lucas *Oxya chinensis* Thunberg	Larve Adult	α -Amylases
Herde and Howe [75]	*Arabidopsis thaliana* L. *Solanum lycopersicum* L.	As a host plants	*Trichoplusia ni* Hübner	Larve	Phase I and II detoxification enzymes Proteinase Serine Proteases Lipase
Oliveira et al. [76]	*Dioclea violacea* Mart. ex Benth.	Lectin (DVL)	*Anagasta kuehniella* Zeller	Larve	Trypsin-Like Chymotrypsin-Like α-amylase Proteases
Jalali et al. [77]	*Cassia angustifolia* Mill. *Trigonella foenum-graecum* L.	Proteinaceous extracts	*Plodia interpunctella* Hübner (Lep.: Pyralidae)	Larve	trypsin
de Oliveira et al. [78]	*Clitoria fairchildiana* R. A. Howard	trypsin inhibitor derived from the cotyledons	*Aedes aegypti* Linnaeus in Hasselquist	Larve	Trypsin-Like A-Amylase Serine Proteases Elastase Chymotrypsin Subtilisin
Li et al. [79]	green bean pods (Gb)	green bean pods and *H. armigera* eggs	*Apolygus lucorum* Meyer-Dür	Adults	Amylase Protease
Shahriari et al. [80]	*Teucrium polium* L.	essential oil	*Ephestia kuehniella* Z. (Lep.: Pyralidae)	Larve	α-amylase Triacylglycerol Lipase General Protease Serine Proteases (Trypsin and Chymotrypsin-Like) Carboxy- And Aminopeptidases
Bezzar-Bendjazia et al. [81]	(Meliaceae) *Azadirachta indica*	Azadirachtin	*Drosophila melanogaster* Meigen	Third instars larvae	α-amylase Chitinase Protease Lipase
Meriño-Cabrera et al. [82]	*Coffea arabica* L.	Leaf extract	*Leucoptera coffeella* Guérin-Méneville (Lepidoptera: Lyonetiidae)	fourth instar larvae	Trypsin Chymotrypsin Cysteine proteases Total protease
Camaroti et al. [83]	*Schinus terebinthifolius* Raddi	saline leaf extract Lectin	*Sitophilus zeamais* Motsch. (maize weevil)	Adults	Protease Amylase
Zou et al. [84]	*Chelidonium majus* L.	alkaloid	*Lymantria dispar*	Third instar larvae	α-amylase Lipase Total Protease
Farhoodi et al. [85]	Iranian wheat cultivars	Seed proteinaceous extracts	*Eurygaster integriceps* Puton	Adults	α-amylase α-glucosidase β-glucosidase proteolytic activities
Cantón and Bonning [86]	corn green bean	Protein Extracts	*Nezara viridula* L.	Second instar nymph	Protease Nuclease Serine Proteases
Fathipour et al. [87]	Brassicaceae two canola cultivars two cabbage cultivars	Host plant	*Plutella xylostella* (L.) (Lepidoptera: Plutellidae)	last-larval instars	proteolytic and amylolytic activities α-glucosidase and β-glucosidases
Zhi et al. [88]	kidney bean	Host plant	*Frankliniella occidentalis* (Pergande)	nymph and adult	α-amylase trypsin tryptase

**Table 2 insects-16-00392-t002:** Summary of research on insect antioxidant enzyme activity in response to plant metabolites and various biotic and abiotic stress factors.

Researcher	Issue	Pest	Investigated Enzymes
Ahmad et al. [109]	Antioxidant enzyme activity is elevated in high-metabolism tissues like Malpighian tubules, hindgut, muscles, and gonads.	*Trichoplusia ni*	superoxide dismutase (SOD) glutathione peroxidase (GPOX) Glutathione-S-transferase (GST) glutathione reductase (GR)
Aucoin et al. [110]	biochemical defenses, against oxidative stress from phototoxins	*Ostrinia nubilalis* Hübner *Manduca sexta* L., *Anaitis plagiata* L.	(SOD) catalase (CAT) (GPOX) (GR).
[111]	effect of mercury as Hg_2_Cl_2_ and HgCl_2_	*Musca domestica* L. *Trichoplusia ni*	SOD CAT GST peroxidase
Wang et al. [112]	Investigation of antioxidant enzyme activities in insect cell lines	*Spodoptera frugiperda* Smith *Trichoplusia ni*	ascorbate peroxidase (APOX) MnSOD CuZnSOD (CAT) GR dehydroascorbic acid reductase
Barbehenn [113]	Investigation of Gut-based antioxidant	*Melanoplus sanguinipes* Fabricius *Aulocara elliotti* Thomas	(SOD) (CAT) (APOX) glutathione transferase peroxidase (GSTPX)
Cervera et al. [114]	Investigation of cadmium toxicity	*Oncopeltus fasciatus* (Dallas)	(CAT) (GR) (GST) thiobarbituric acid reactive substances (TBARS)
Lijun et al. [115]	The effect of different concentrations of cadmium (Cd^2+^)	*Oxya chinensis* Thunberg (Orthoptera: Acridoidae)	(SOD) (CAT) guaiacol peroxidase (GPx)
Krishnan and Kodrík [116]	The effect of host plant and artificial diet on enzyme activity	*Spodoptera littoralis*	(SOD) (CAT) (APOX) (GSTpx)
Mittapalli et al. [117]	Insect antioxidant activity in interaction with host plant	*Mayetiola destructor*	phospholipid glutathione peroxidases (MdesPHGPX-1 and MdesPHGPX-2) catalases (MdesCAT-1 and MdesCAT-2) (MdesSOD-1 and MdesSOD-2)
Hyršl et al. [118]	boric acid-induced oxidative stress	*Galleria mellonella* (L.)	(SOD) (CAT) (GST) (GPx)
Wang et al. [119]	ultraviolet-A stress	*Helicoverpa armigera*	(Cu/ZnSOD) (CAT) (GPX)
Büyükgüzel et al. [120]	Effect of boric acid on antioxidant enzyme activity	*Galleria mellonella* L.	CAT SOD GST GP
Büyükgüzel et al. [120]	Effect of cadmium	*Galleria mellonella*	lipid peroxidation (MDA) (SOD) (CAT)
Jena et al. [121]	Effect of High temperatures	*Antheraea mylitta* L.	(SOD) (CAT) (GST) ascorbic acid (ASA)
Büyükgüzel and Kalender [122]	Penicillin-Induced Oxidative Stress	*Galleria mellonella*	[SOD] [CAT] [GST] [GPx]
Dere et al. [123]	Effect of azadirachtin	*Galleria mellonella* (Lepidoptera: Pyralidae)	GST
Wu and Yi [124]	effects of chromium (Cr) and lead (Pb)	*Galleria mellonella*	phenoloxidase, PO
Li et al. [125]	Effect of Entomopathogenic nematodes (EPNs) Steinernema and Heterorhabditis	*Tenebrio molitor* L.	(SODs) peroxidases (PODs) (CATs) tyrosinase (TYR) acetylcholinesterase (AChE) carboxylesterase (CarE) (GSTs)
Ali et al. [126]	Thermal stress	*Mythimna separata* Walker (Lepidoptera: Noctuidae)	(SOD) (CAT) (POX) (GSTs) total antioxidant capacity (T-AOC)
Ali et al. [127]	Influence of UV-A radiation	*Mythimna separata* (Lepidoptera: Noctuidae)	SOD CAT POX GST
Dixit et al. [128]	Effect of cotton phenolics	*Helicoverpa armigera* *Spodoptera litura*	lactate dehydrogenase GST
Karthi et al. [129]	Effect of *Aspergillus flavus*	*Spodoptera litura*	Phenoloxidase CAT POX SOD Lipid peroxidase Acid phosphatase Alkaline phosphatase
Islam et al. [130]	heavy metals stress (Cd, Pb, Mn, and Zn)	*Antheraea assamensis* Helfer (Lepidoptera: Saturniidae)	GST SOD CAT
Ramadan et al. [131]	Effects of synthesized silver and chitosan nanoparticles	*Musca domestica*	SOD CAT GPx GST
Miao et al. [132]	Heat shock	*Liposcelis bostrychophila* Badonnel	(SOD) (CAT) (POD) (GST) (MDA)
Farahani et al. [133]	impact of temperature extremes, starvation, and parasitism by *Habroacon hebetor* wasps.	*Ectomyelois ceratoniae* Zeller	(SOD) (CAT)
Li et al. [134]	Infection by the fungus *Ascosphaera apis*	*Apis mellifera* L.	SOD CAT GST
Muhammad et al. [135]	biological impacts linked to dietary exposure to CuO and ZnO nanoparticles	*Bombyx mori*	SOD GST CAT
Chamani et al. [8]	Effect of Zn, Cu, and Fe nanoparticles and Urea	*Schizaphis graminum* Rondani	SOD CAT POX
Janković-Tomanić et al. [136]	Impact of elevated levels of zearalenone	*Tenebrio molitor* L. (Coleoptera: Tenebrionidae)	(SOD) (GST)
Ma et al. [137]	Impacts of acute and chronic exposure to chromium stress	*Orthetrum albistylum* Selys	SOD CAT

## Data Availability

The data presented in this study are available on request from the corresponding author.

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
