# Peer review of "From Digestion to Detoxification: Exploring Plant Metabolite Impacts on Insect Enzyme Systems for Enhanced Pest Control"

_insects, 2025, doi:10.3390/insects16040392_

Round 1
Reviewer 1 Report
Comments and Suggestions for Authors
The file with the corrections is attached.

The file with the corrections is attached.
Reviewer 2 Report
Comments and Suggestions for Authors
This review outlines the relationship between plants and insects and the various chemical interactions that have evolved between them. The authors do an excellent job of outlining some of the various interactions in the arms race between plants and insects and how this can be exploited to develop more bio-friendly pest control measures. While the authors give a good number of lists for plant metabolites, insect enzyme systems, ROS/antioxidants, and abiotic factors, there are several majors concerns I have with the overall manuscript.
1. The images that are used are not intuitive. There are many words and arrows, but the over flow of the images does not necessarily tell "a story" or convey meaning. Also, several of the images are very cartoon-ish for a professional manuscript.
2. The list of interactions, enzymes, and plant/insect systems is very incomplete. For example, there is only one mention of glucosinolates in the article, but there is a whole body of work on how certain "specialist species" such as Pieris rapae can easily break down glucosinolates using specific hindgut enzymes. This is just one example. The authors don't go into detail about any of the systems. Thus, this review is hardly exhaustive.
3. Overall, the paper is vague and redundant. There are many paragraphs that can be omitted due to content already said. I highly suggest getting into details with several of these systems and really showing the reader the in-depth workings of how the enzymes systems have evolved in insects to breakdown---either enzymatically or non-enzymatically various plant metabolites. But, be specific. Perhaps the scope of the paper is too broad for the number of topics that are covered or the paper needs to be twice as long with more than double the number of references.
4. I am grateful to see that the authors are suggesting more bio-friendly methods for pest management by taking advantage of biological systems. However, can you give any specific examples of current pest control measures that are using insect enzyme inhibitors (ie. enzymes that specifically target plant metabolites). Are there other abiotic factors that can be used in conjunction with these? The paper briefly mentions these, but does not go into detail.
Reviewer 3 Report
Comments and Suggestions for Authors
The review article explores the interactions between plant metabolites and insect enzyme systems, emphasizing their roles in digestion, detoxification, and pest control strategies. It discusses how plant-derived compounds influence insect digestive and antioxidant enzymes and their potential applications in sustainable pest management. Overall, the manuscript presents an interesting and timely review of the biochemical interactions between plant metabolites and insect enzymes. The review is generally well-structured, providing extensive background information and covering various aspects of enzyme regulation. However, several areas require improvement before the manuscript can be considered for publication.
Major Concerns:
Mechanistic Depth: The manuscript describes what happens when insects encounter plant metabolites (e.g., enzyme inhibition, oxidative stress) but does not go deeply into the underlying molecular mechanisms. For example:
How do different classes of plant metabolites interact with insect enzymes at the molecular level?
What are the latest advances in understanding how insects evolve resistance to plant enzyme inhibitors?
Repetition and Organization: Some sections contain overlapping content, leading to redundancy. For example, the role of antioxidant enzymes in detoxifying plant metabolites is discussed in both Section 2.2 and Section 3.2, but their specific focus is not clearly differentiated. Additionally, transitions between sections could be smoother to improve readability.
Interplay Between Digestive and Antioxidant Enzymes: The manuscript notes ROS production during digestion (Section 4) but does not explore its connection to digestive enzyme inhibition—such as how ROS might damage protease active sites and impact enzyme function. A discussion linking these aspects would provide a more comprehensive perspective.
Minor Concerns
Figure Legends: Some figures lack sufficient explanation. Ensure that all figure legends provide clear and complete descriptions.
Additional Citations: To strengthen the discussion on insect digestive enzyme plasticity and detoxification mechanisms, I recommend citing the following relevant studies:
- Physiological adaptations to sugar-mimic alkaloids: Insights from Bombyx mori for long-term adaption and short-term response. (DOI: 10.1002/ece3.6574)
- Host-specific gene expression as a tool for introduction success in Naupactus parthenogenetic weevils. (DOI: 10.1371/journal.pone.0248202)
- Expression plasticity and evolutionary changes extensively shape the sugar-mimic alkaloid adaptation of nondigestive glucosidase in lepidopteran mulberry-specialist insects. (DOI: 10.1111/mec.14720)
- Molecular mechanisms of insect adaptation to plant secondary compounds. (DOI: 10.1016/j.cois.2015.02.004)
- Gut bacteria of lepidopteran herbivores facilitate digestion of plant toxins. (DOI: 10.1073/pnas.2412165121)
These papers provide additional insights into evolutionary adaptations in insect metabolism and the co-evolutionary arms race between plants and herbivorous insects.
Section 2.1, Page 4: "Insects have a variety of digestive enzymes such as proteases, lipases, amylases, sucrases and trehalose, which they secrete in their guts." "Trehalose" is listed as a digestive enzyme, which is incorrect. Trehalose is a disaccharide sugar, not an enzyme. The correct term should be "trehalase."
"Enzyme modulating properties" should be "enzyme-modulating properties."
